# Phytochemical Investigation and Biological Studies on Selected *Searsia* Species

**DOI:** 10.3390/plants11202793

**Published:** 2022-10-21

**Authors:** Mkhuseli Koki, Masande Yalo, Masixole Makhaba, Ndikho Nako, Fanie Rautenbach, Jelili A. Badmus, Jeanine Marnewick, Ahmed A. Hussein, Wilfred T. Mabusela

**Affiliations:** 1Department of Chemistry, University of the Western Cape, Private Bag X17, Bellville 7538, South Africa; 2Department of Pharmacology, Faculty of Health Sciences, University of Free State, Bloemfontein 9300, South Africa; 3Chemistry Department, Cape Peninsula University of Technology, Symphony Rd., Bellville 7535, South Africa; 4Applied Microbial and Health Biotechnology Institute, Cape Peninsula University of Technology, Symphony Rd., Bellville 7535, South Africa

**Keywords:** *Searsia* *genus*, amentoflavone, moronic acid, alpha-glucosidase, alpha-amylase, anti-diabetic activity, enzyme inhibition, oxidative stress

## Abstract

*Searsia* is the more recent name for the genus *Rhus*, which contains over 250 individual species of flowering plants in the family *Anacardiaceae*. Several *Searsia* species are used in folk medicine and have been reported to exhibit various biological activities. Although known to exhibit different terpenoids and flavonoids, the chemistry of the *Searsia* genus is not comprehensively studied due to the structural complexity of the compounds. In this study, the extraction, isolation, and identification of secondary metabolites from three *Searsia* species (*Searsia glauca, S. lucida*, and *S. laevigata*) were conducted using chromatographic and spectroscopic techniques and afforded five known terpenes, viz., moronic acid (**1**), 21β-hydroxylolean-12-en-3-one (**2**), lupeol (**11**), α-amyrin (**9**), and β-amyrin (**10**), in addition to six known flavonoids, myricetin-3-*O-*β-galactopyranoside (**3**), rutin (**4**), quercetin (**5**), apigenin (**6**), amentoflavone (**7**), and quercetin-3-*O-*β-glucoside (**8**). The structural elucidation of the isolated compounds was determined based on NMR (1D and 2D) and comparison with the data in the literature. Biological assays, such as antioxidant and enzyme inhibition activity assays, were conducted on the plant extracts and the isolated compounds. The antioxidant capacities of hexane, dichloromethane, ethyl acetate, methanol, and butanol main extracts were investigated using ferric ion reducing power (FRAP), oxygen radical absorbance capacity (ORAC), and Trolox equivalent antioxidant capacity (TEAC) assays. The results showed high antioxidant activities for methanol and butanol extracts of the three plants. The isolated compounds were tested against alpha-glucosidase and alpha-amylase, and the results showed the potent activity of moronic acid (**C1**) (IC_50_ 10.62 ± 0.89 and 20.08 ± 0.56 µg/mL, respectively) and amentoflavone (**C7**) (IC_50_ 5.57 ± 1.12 µg/mL and 19.84 ± 1.33 µg/mL, respectively). Isolated compounds of and biological assays for *S. glauca, S. lucida*, and *S. laevigata* are reported for the first time.

## 1. Introduction

Diabetes is rapidly emerging as a global health problem that threatens to reach epidemic levels by the year 2030 [1]. It is another serious chronic metabolic disorder characterized by chronic hyperglycemia. Hyperglycemia results from abnormal metabolism of carbohydrates, lipids, and protein. Diabetes mellitus is characterized by hyperglycemia resulting from defects in insulin secretion, action, or both [2]. The condition is commonly classified into type 1 and type 2. Type 1 diabetes is usually referred to as insulin-dependent, since it results from the failure of the pancreatic cells to secrete insulin and its complications are managed by injections of exogenous insulin. Type 2 diabetes is the most common type of diabetes; it is a metabolic disorder, with multiple etiologies, characterized by carbohydrate, lipid, and protein metabolic disorders that include defects in insulin secretion, with a major contribution to insulin resistance [3]. The management of the disease is important for its control, which involves lowering the postprandial increase in blood glucose levels by inhibiting the enzymes alpha-amylase and alpha-glucosidase. Alpha-glucosidase and alpha-amylase are the key enzymes involved in the digestion of carbohydrates and starch into simple sugars which leads to an increase in blood glucose [4,5]. Alpha amylase degrades complex dietary carbohydrates to oligosaccharides and disaccharides, which are further converted into monosaccharides by alpha glucosidase. These enzymes are responsible for the hydrolysis of carbohydrates to simple sugars, such as glucose [6]. Inhibition of alpha-glucosidase retards the digestion of carbohydrates, resulting in a reduction in the rate of glucose absorption [7,8]. Acarbose is being used as an alpha-glucosidase inhibitor, but it has side effects, such as bloating, flatulence, and diarrhea [9]. Natural products may be feasible alternative remedies for the treatment of diabetes or be complementary to currently used treatments [10]. Medicinal plants have been used worldwide for the therapy of diabetes type 2. These are plants that have trifoliate leaves with small flowers that eventually produce fruits and they belong to the category of drupes. Some compounds from the *Searsia* species show anti-HIV-1 activities [11], antioxidant properties, and anti-diabetic activities that can be explored for the development of novel anti-HIV agents [12,13]. Previous studies conducted on *Searsia* plants have investigated their antioxidant activities and shown the potential for commercial development of products obtained from several species [12]. *Searsia* species can grow in non-agriculturally viable regions without necessarily competing with food production in terms of land use and have been used by indigenous cultures for medicinal and other purposes [14]. It was reported that *R. chondroloma* showed high levels of flavonoids and tannins, as well as terpenes and steroids [15]. The antioxidant activity of *Rhus chondroloma* ethanolic extract is probably linked to the high flavonoid and phenolic contents, whereas the anti-lipase effects could be associated with the latter compounds but possibly with other metabolites, such as steroids [15]. Several plants are traditionally used to treat epilepsy and convulsions, including some *Searsia* species, such as *Searsia chirindensis*, *S. dentate*, *S. natalensis*, and *S. pyroides* [16]. Extracts from *Searsia* species, such as *S. coriaria*, *S. chirindensis*, and *S. verniciflua*, have been reported to exhibit significant in vitro and in vivo hypoglycemic activities [17,18,19]. The leaves and fruits of *S. coriaria* have been reported to exert defensive and beneficial effects on a wide set of diseases, including, but not limited to, diabetes mellitus, cancer, stroke, oral diseases, inflammation, diarrhea, and dysentery [20]. The essential oil from *R. lancea* is reported to have potent antioxidant activity [21]. Although *Searsia* species have been reported to express a wide range of biological activities, there is still a lack of comprehensive studies on their phytochemical analysis. In addition, there are no ethnomedicinal reports historically conducted on two of the three *Searsia* species, viz., *S. lucida* and *S**. laevigata*, investigated in this study. The ethanol leaf extract of *S. glauca* has been studied and shown to act as a possible antagonist of *N*-methyl-D-aspartate (NMDA)-type glutamate receptors [22]. Antioxidant and enzyme inhibition evaluations of the extracts and/or compounds from three *Searsia* species, viz., *S. glauca*, *S. lucida*, and *S. laevigata*, are reported for the first time in this study.

## 2. Results and Discussion

### 2.1. Chemical Characterization

The chemical structures of the known compounds **1**–**11** (Figure 1) were elucidated based on 1D (^1^H, ^13^C, and DEPT 135) and 2D-NMR experiments (HSQC and HMBC) and confirmed by comparison of their spectroscopic data with the data available in the literature.

#### Structural Elucidation of the Isolated Compounds

The chemical structure of compound **C1** was determined by comparing (^1^H and ^13^C) spectral data (See Appendix A) with those for moronic acid, as reported in the literature [22,23]. Compound **C1** was previously isolated from *S. Javanica* [24]. From the ^1^H and ^13^C NMR spectra, compound **C2** was confirmed to be a 12-oleanene type triterpene with a secondary hydroxyl group and a keto group. After comparing all the experimental spectroscopic data with previously reported data, compound **C2** was identified as 21β-hydroxylolean-12-en-3-one, and this compound has been isolated from *Hippocratea excelsa* [25]. Compound **C3** was isolated from butanol main extract of *S. glauca*. The ^1^H and ^13^C NMR data (See Appendix A) for compound **C3** agreed with the data previously reported in the literature for myricetin-3-*O*-β-galactopyranoside identification [26]. Compound **C4** was identified as rutin [27]; this compound was previously reported from *R. natalensis* [13]. Compound **C5** was identified as quercetin [28]. Compound **C6** was identified as apigenin [29,30]. The ^13^C and ^1^H NMR data (See Appendix A) for compound **C7** were compared with those for an amentoflavone isolated from *Podocarpus nakaii* [31]. This compound was isolated from *S. succedanea*, *S. retinorrhea*, and *S. pyroides* [32,33,34]. Compound **C8** was identified as quercetin 3-*O*-β-glucoside [35]. The ^1^H and ^13^C NMR data for compound **C10** were compared with previously reported data for β-amyrin [36]. An inseparable mixture of three isomeric triterpenoids (**C9**, **C10**, and **C11**) was isolated from dichloromethane main extract. The ^1^H and ^13^C NMR data for this mixture indicated compound **C11** as lupeol; the presence of β-amyrin (**C10**) and α-amyrin (**C9**) in the mixture were confirmed after comparing the experimental data and previously reported data [39/36,40/37].

### 2.2. Biological Activity

#### 2.2.1. Evaluating Oxygen Radical Absorbance Capacity (ORAC) Activities of the *S. glauca*, *S. lucida*, and *S. laevigata* Extracts

The regression equation between net AUC and Trolox concentration was established, and ORAC values were reported as µM TE/g of plant extract, using the standard curve determined previously (Figure 2). Ethyl acetate and butanol main extracts of *S. glauca* exhibited the highest ORAC values of 4574.93 ± 109.12 and 5653.36 ± 328.66 µM TE/g (Table 1), respectively. The ethyl acetate, methanol, and butanol main extracts of *S. lucida* showed the highest ORAC values of 4010.56 ± 73.52, 5793.45 ± 27.30, and 4198.42 ± 166.53 µM TE/g, respectively. Hexane and dichloromethane main extracts showed the lowest ORAC values of 1013.65 ± 12.26 and 1048.15 ± 35.35 µM TE/g (Table 1), respectively.

#### 2.2.2. Evaluating Ferric Ion Reducing Power (FRAP) Activities of the *S. glauca*, *S. lucida*, and *S. laevigata* Extracts

The FRAP values were generated by comparing the absorbance changes in the test mixtures with those obtained from increasing concentrations of Fe^3+^ and were expressed as µM AAE/g of the samples (Figure 3). The FRAP activities of the main extracts from *S. glauca*, *S. lucida*, and *S. laevigata* are presented in Table 2. The methanol extract, ethyl acetate, and butanol main extracts of *S. lucida* exhibited high FRAP values of 1038.39 ± 80.41 µM AAE/g, 137.24 ± 16.54 µM AAE/g, and 680.01 ± 96.34 µM AAE/g, respectively. Hexane and dichloromethane main extracts showed low FRAP values of 12.99 ± 1.82 µM AAE/g and 40.69 ± 2.11 µM AAE/g, respectively. The FRAP values obtained from methanol, ethyl acetate, and butanol main extracts of *S. laevigata* showed significant activity, i.e., 411.58 ± 79.73, 86.28 ± 9.16, and 977.88 ± 71.24 µM AAE/g, respectively. It was observed that butanol, ethyl acetate, and dichloromethane main extracts from *S. glauca* showed high FRAP values, i.e., 1008.67 ± 41.65 µM AAE/g, 557.09 ± 39.41 µM AAE/g, and 431.62 ± 39.34 µM AAE/g, respectively. 

#### 2.2.3. Evaluating Trolox Equivalent Antioxidant Capacity (ABTS/TEAC) Activities of the *S. glauca*, *S. lucida*, and *S. laevigata* Extracts

The antioxidant power of the extracts was examined using the ABTS radical decolorization assay, which measures the relative antioxidant ability to scavenge the radical ABTS^+^ (2.2′-azinobis (3-ethylbenzothiazoline-6-sulfonic acid)), a blue-green chromophore with maximum absorption at 734 nm that decreases in its intensity in the presence of antioxidants. Antioxidants can neutralize the radical cation ABTS^+^, generated from ABTS, by direct reduction via electron donation, or the balance of these two mechanisms is determined by the antioxidant structure and pH of the medium [37,38,39]. The TEAC values of various extracts of *S. lucida*, *S. laevigata*, and *S. glauca* are shown in Figure 4 and Table 2. The methanolic extract of *S. lucida* exhibited the highest TEAC value of 1512.08 ± 0.06 (µM TE/g), and butanol extract of *S. glauca* indicated the highest TEAC activity of 1438.63 ± 13.53 (µM TE/g). The ethyl acetate of *S. glauca* showed the highest TEAC value of 1095.42 ± 28.42 (µM TE/g). Hexane and dichloromethane extracts of *S. lucida* showed low TEAC values of 20.26 ± 4.42 and 52.20 ± 2.82 (µM TE/g), respectively. Generally, extracts with high contents of phenolics exhibited high radical scavenging and antioxidant activities [40,41,42].

#### 2.2.4. Evaluation of In Vitro Activity against Carbohydrate Digestive Enzymes (Alpha Glucosidase and Alpha Amylase)

The extracts of *Searsia* species were screened to determine their inhibition activities with respect to enzymes relevant to the management of diabetes. The extracts were tested at 2.0 mg/mL. The results for both enzymes are shown in Table 3—extracts with 25–50% inhibitory effect on α-glucosidase (low potency), 50–75% (moderate potency), and 75–100% (high potency) [43]. The enzyme inhibition of *S. glauca* extracts was between 41.2 and 50.08% for α-amylase and between 55.76 and 85% for α-glucosidase; *S. lucida* extracts were between 24.95 and 60.88% for α-amylase and between 49.28 and 93.54% for α-glucosidase; *S. laevigata* extracts showed an enzyme inhibition between 40.99 and 59.72% for α-amylase and for alpha glucosidase between 68.28 and 90.10%.

The isolated compounds from the three plants were tested against alpha glucosidase and alpha amylase at 125.0 µg/mL to determine their inhibitory activities. The compounds that showed activity during screening were tested further to determine their IC_50_ values at various concentrations (125.0, 62.5, 31.25, 15.63, 7.82, 3.91, and 1.95 µg/mL). It was only compounds **C1** and **C7** that showed activity at 125.0 µg/mL, and these were tested against the two digestive enzymes to examine their IC_50_ values, as shown in Table 3. Compounds **C1** and **C7** showed more potent inhibition values for alpha glucosidase and alpha amylase: IC_50_ 10.57 ± 2.02 and 20.08 ± 0.98 µg/mL; IC_50_ 5.57 ± 1.17 and 19.84 ± 1.03 µg/mL, respectively. Compound **C1** showed high potency against alpha glucosidase compared with the positive control (quercetin), with an IC_50_ value of 105.41 ± 2.30 µg/mL, and **C7** showed the highest potency, with an IC_50_ value of 5.57 µg/mL. 

## 3. Materials and Methods

### 3.1. Plant Materials

The aerial parts of *Searsia glauca*, *Searsia lucida*, and *Searsia laevigata* were collected in February 2013 from the Cape Nature Reserve at UWC in Cape Town, South Africa. Species were identified and confirmed by Mr. Franz Wertz from the University of the Western Cape Herbarium. Samples were dried, powdered, and stored in darkness in a refrigerator until further use.

### 3.2. Equipment and Chemical Reagents

The 1D (^1^H, ^13^C, and DEPT-135) and 2D NMR (COSY, HSQC, and HMBC) spectra were recorded with an Avance 400 MHz NMR spectrometer (Bruker, Rheinstetten, Germany) at 400 (proton, ^1^H) and 100 (carbon, ^13^C) MHz. Chemical shifts were reported in parts per million (ppm) and coupling constants (*J*) in Hz. The ^1^H and ^13^C NMR values were relative to the internal standard TMS and were acquired in CD_3_OD, CDCl_3_, or DMSO-d_6_. HRESI-MS were obtained on a Waters Synapt G2 mass spectrometer (Cone Voltage 15 V), which was operated in the negative and/or positive ion mode using direct injection. ATR-FTIR (PerkinElmer Spectrum 100; Llantrisant, Wales, UK) was used, at a transmission mode of 400–4000 cm^−1^. Column chromatography was performed using Sephadex (LH-20; Sigma-Aldrich, St. Louis, MO, USA) and normal-phase silica gel 60 (70–230-mesh ASTM; Merck, Kenilworth, NJ, USA). TLC was performed on silica gel aluminum sheets (Silica gel 60 F254; Merck) to monitor the fractions. Visualization was achieved with 10% H_2_SO_4_ and detection with vanillin sulfuric acid reagent and heating to 105 °C.

### 3.3. Extraction and Fractionation of the Plant Material

#### 3.3.1. *Searsia glauca*

Leaves and flowers of *Searsia glauca* (300.05 g) were air-dried at room temperature, blended, and extracted (2.5 L × 2) with 80% methanol for 48 h. The methanol extract was evaporated at 40 °C using a rotary evaporator. The methanol extract was partitioned with hexane (15.42 g), dichloromethane (10.14 g), ethyl acetate (8.25 g), and butanol (6.12 g). The extracts were loaded on the chromatography column with silica gel and eluted with hexane: ethyl acetate (100:0; 95:05; 90:10; 80:20; 70:30; 60:40; 50:50; 40:60; 20:80; 0:100). The collected fractions were concentrated using a rotary evaporator and profiled on TLC. All the fractions with similar profiles were combined and reloaded into the column (CC) until the pure compound was obtained or further purified using prep HPLC-MS. Fraction B (1.27 g) was reloaded in the chromatography column with silica gel (0.063–0.2 mm; 70–230-mesh ASTM) and eluted with hexane: ethyl acetate (9:1; 7:3), from which 100 mL of each fraction was collected and evaporated using a rotary evaporator. Fractions with similar profiles were combined. The chromatography of sub-fraction A1 (340.23 g) on silica gel (hexane: ethyl acetate; 7:3) resulted in the isolation of pure compounds **1** (82 mg) and **2** (8 mg). The compounds were crystalized using methanol. 

#### 3.3.2. *Searsia*
*lucida*

Leaves and flowers of *Searsia lucida* (280.12 g) were air-dried at room temperature, blended, and extracted (2.5 L × 2) with 80% methanol for 48 h. The methanol extract was evaporated at 40 °C using a rotary evaporator. The methanol extract was partitioned with hexane (16.42 g), dichloromethane (12.14 g), ethyl acetate (10.25 g), and butanol (6.12 g). The crude extracts were each subjected to column chromatography (CC) for the isolation of pure compounds. Crude extracts were eluted with hexane: ethyl acetate and dichloromethane: ethyl acetate (100:0; 80:20; 60:40; 50:50; 20:80; 0:100) and washed with 100% methanol. Fractions with similar profiles after TLC profiling were combined and further purified on CC or preparative HPLC-MS. Fraction FH6 (250.36 mg) was loaded on the silica column and eluted with hexane: ethyl acetate (8:2). After profiling the fractions with a TLC plate, a pure compound (moronic acid) (40.52 mg) was obtained. During fractionation, dichloromethane and ethyl acetate extracts were combined due to their similar TLC profiles. Fraction FDE-A (150.12 mg) was chromatographed on silica gel and eluted with a mobile phase of dichloromethane: ethyl acetate (7:3) and afforded compound **6** as a yellow powder (15.81 mg). Fraction FDE-B (300.56 mg) was chromatographed on silica gel and eluted with dichloromethane: ethyl acetate (1:1). Sub-fractions from 12–17 precipitated, and yellow porous powder was observed. The yellow powder was further rinsed with methanol. FDE-B afforded two compounds, **6** (6.12 mg) and **7** (110.21 mg). 

#### 3.3.3. *Searsia*
*laevigata*

Leaves and flowers of *Searsia laevigata* (200.10 g) were air-dried at room temperature, blended, and extracted (2.5 L × 2) with 80% methanol for 48 h. The methanol extract was evaporated at 40 °C using a rotary evaporator. The methanol extract was partitioned with hexane (10.54 g), dichloromethane (7.22 g), ethyl acetate (5.18 g), and butanol (4.12 g). All the crude extracts were profiled using TLC, and each was subjected to repeated column chromatography (CC) elution with hexane: ethyl acetate, dichloromethane, and ethyl acetate: methanol, depending on the polarity of the extract. Fraction A, from a dichloromethane extract, was subjected to column chromatography over silica gel. The fraction was eluted with hexane: ethyl acetate (8:2). The sub-fractions with similar profiles were combined and re-chromatographed over flash chromatography silica gel, followed by elution with 100% dichloromethane, and the TLC showed a single spot. The amount of the mixtures was 22.23 mg (compounds **9**–**11**). Fraction A from butanol extract was subjected to chromatography silica gel and eluted with ethyl acetate: methanol (8:2). After five repeated runs over the silica gel, sub-fractions with similar profiles were further purified on preparative HPLC-MS and afforded compound **8** (8.03 mg) (quercetin-3-O-β-glucoside).

### 3.4. General Experimental Procedure for Bioassays

#### 3.4.1. Reagents

Standards (purity > 99.0%) for antioxidants, Trolox (6-hydroxyl-2,5,7,8-tetramethylchroman-2-carboxylic acid; Saarchem, Mumbai, India), and other reagents, including ABTS (2,2′-azino-bis (3-ethylbenzo thiazoline-6-sulfonic acid) diammonium salt (Sigma), potassium-peroxodisulphate (Merck), Ethanol (Saarchem), fluorescein sodium salt (Sigma), AAPH (2,2′-azobis (2-methylpropionamidine) dihydrochloride (Aldrich), Sodium dihydrogen orthophosphate dihydrate (Na_2_HPO_4_.2H_2_O, Merck), PCA (perchloric acid, Saarchem), TPTZ (2,4,6-tri[2-pyridyl]-*s*-triazine), iron (III) chloride hexahydrate, and copper sulphate, were purchased from Sigma-Aldrich, Inc. (St. Louis, MO, USA). L-Ascorbic acid was procured from Sigma Aldrich, Cape Town, South Africa. Antioxidant assays, including FRAP and TEAC assays, were performed using a Multiskan spectrum plate reader, while an automated ORAC assay was performed with a Floroskan spectrum plate reader. For the enzyme inhibition studies, α-glucosidase (*Saccharomyces*
*cerevisiae*), α-amylase (procaine pancreas) and 3, 5, di-nitro salicylic acid (DNS), p-nitro-phenyl-α-D-glucopyranoside (*p*-NPG), sodium carbonate (Na_2_CO_3_), soluble starch, sodium dihydrogen phosphate, and disodium hydrogen phosphate were purchased from Sigma-Aldrich, South Africa.

#### 3.4.2. Ferric Ion Reducing Antioxidant Power (FRAP) Assay

Preparation of the FRAP reagent in a 50 mL conical flask was composed of 30 mL Acetate buffer (300 mM, pH 3.6; 1.627 g Sodium acetate + 16 mL Glacial acetic acid and made up to 1 L with distilled water) + 3 mL TPTZ 10 mM solution (0.0093 g TPTZ and 3 mL of 40 mM HCl in 15 mL flask) + 3 mL FeCl_3_ solution (0.054 g FeCl_3_.6H_2_O and 10 mL distilled water) and 6.6 mL of distilled water. A control solution was prepared by dissolving 0.00352 g of Ascorbic acid in 50 mL, diluted to the mark with distilled water. A standard curve was prepared as per the method described by Benzie and Strain, 1996. L-Ascorbic acid (Sigma Aldrich, Cape Town, South Africa) was used as a standard, with concentrations varying between 0 and 1000 µM. Dilutions were made for the extracts that were highly concentrated, and dilution factors were noted and considered in calculations for the relevant extracts. In a 96-well clear microplate, 10 µL of the stock solutions of the extracts (hexane, dichloromethane, ethyl acetate, methanol, and butanol) of three plants were mixed with a 300 µL FRAP reagent. The plate was read at a wavelength of 593 nm in a multiplate reader. The calibration range was made up between 0 mg/L (blank) and 500 mg/L of L-ascorbic acid for analytical measurement.. The results were reported as μmol ascorbic acid equivalents per mmol (μmolAAE/mmol) of the dry-weight test samples.

#### 3.4.3. Trolox Equivalent Absorbance Capacity (TEAC) Assay

The total antioxidant activities of the extracts were determined by previously described methods [44,45]. The ABTS mix solution was diluted with ethanol to read a start-up absorbance of approximately 2.0 (±0.1). The test extracts were prepared by dissolving 1 mg of an extract in 1 mL of methanol, allowed to react with 300 µL ABTS solution in the dark at room temperature for 30 min. The absorbance was measured at 734 nm at 25 °C in a multiplate reader. The results were expressed as µM Trolox equivalents per milligram dry weight (µM TE/g) of the test samples.

#### 3.4.4. Automated Oxygen Radical Absorbance Capacity (ORAC) Assay

ORAC was determined according to a previously described method [46], with some modifications [47]. Fluorescein was used as the fluorescent probe. The loss of fluorescence of fluorescein was an indication of the extent of oxidation through reaction with peroxyl or hydroxyl radicals. The protective effect of an antioxidant was measured by assessing the fluorescence area under the curve plot relative to that of a blank in which no antioxidant was present. The ORAC value was calculated by dividing the area under the sample curve by the area under the Trolox curve, with both areas being corrected by subtracting the area under the blank curve.

#### 3.4.5. Alpha Glucosidase Inhibitory Activity

The alpha glucosidase inhibitory activities of the extracts were determined according to the standard method, with minor modifications [48]. In a 96-well plate, a reaction mixture containing 50 μL phosphate buffer (100 mM, pH = 6.8), 10 μL alpha glucosidase (1 U/mL), and 20 μL of extract at 2.0 mg/mL was pre-incubated at 37 °C for 15 min. Then, 20 μL P-NPG (5 mM) was added as a substrate and incubated further at 37 °C for 20 min. The reaction was stopped by adding 50 μL Na_2_CO_3_ (0.1 M). The absorbance of the released p-nitrophenol was measured at 405 nm using a Multiplate Reader. The results were reported as half-maximal inhibitory (IC_50_) concentrations in μM. Quercetin was used as a control. Each extract was tested in triplicate.
(1)$$\mathrm{Inhibitory}\;\mathrm{activity}\;(\%)=1\;-\;\frac{As}{Ac}\times 100\;$$
where *As* is the absorbance in the presence of the test substance and *Ac* is the absorbance of the control.

#### 3.4.6. Alpha Amylase Inhibitory Assay

The alpha amylase inhibitory assays of the extracts were carried out according to the standard method, with minor modifications [49]. In a 96-well plate, the reaction mixture, containing 50 μL phosphate buffer (100 mM, pH = 6.8), 10 μL alpha amylase (2 U/mL), and 20 μL of plant extracts at 2.0 mg/mL, was preincubated at 37°C for 20 min. Then, 20 μL of 1% soluble starch (100 mM phosphate buffer pH 6.8) was added as a substrate and incubated further at 37 °C for 30 min; 100 μL of the DNS colour reagent was then added and boiled for 10 min. The absorbance of the resulting mixture was measured at 540 nm using a multiplate reader. Acarbose was used as a control. The extracts or isolated compounds were measured in triplicate. The results were reported as half-maximal inhibitory (IC_50_) concentrations in μM. The results for both alpha amylase and alpha glucosidase were calculated using equation 1.

#### 3.4.7. Statistical Analysis

All the measurements were repeated three times, and the IC_50_ values were calculated using GraphPad Prism 8 version 8.4.3 (Graph pad software, Inc., La Jolla, CA, USA) statistical software. The data presented are means ± SDs obtained from 96-well plate readers for all in vitro experiments.

## 4. Conclusions

Phytochemical investigation of *S. glauca, S. lucida,* and S. *laevigata* led to the isolation of seven known terpenes and six flavonoids. The isolated compounds are reported for the first time from these plant species. The findings of this study show that polar solvent main extracts may be a potential source of secondary metabolites with high antioxidant capacities. Consequently, these plants can be used as potent sources of antioxidants. The enzyme inhibition exhibited by the main extracts and two compounds validates the potential of these plant species for ethnomedicinal use in folk medicine.

## Figures and Tables

**Figure 1 plants-11-02793-f001:**
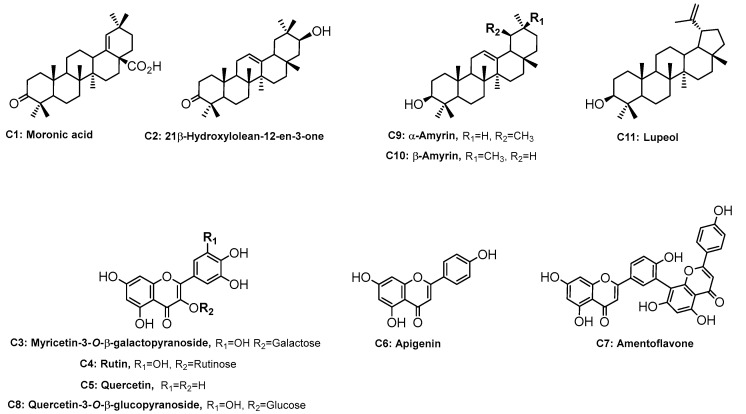
Chemical structures of the isolated compounds (**1**–**11**) from *S. gluaca, S. lucida*, and *S. laevigata*.

**Figure 2 plants-11-02793-f002:**
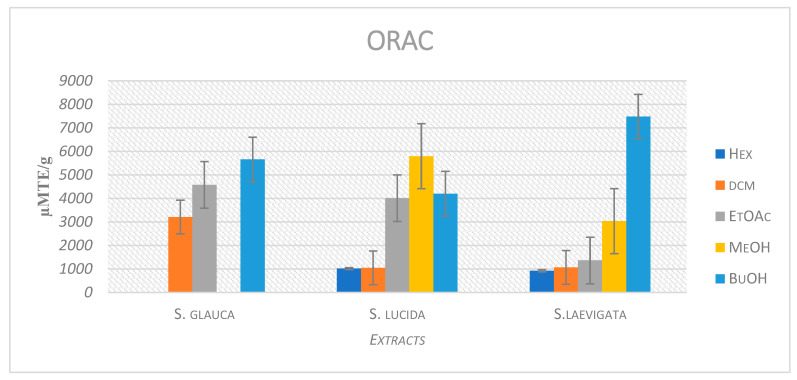
Bar chart of ORAC values of extracts from *S. lucida*, *S. laevigata*, and *S. glauca*.

**Figure 3 plants-11-02793-f003:**
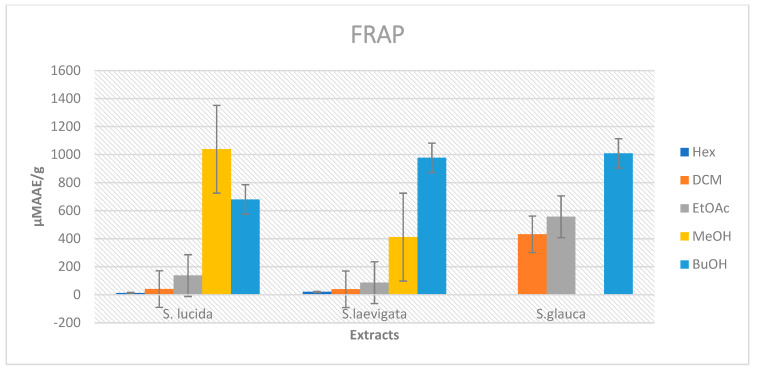
Bar chart of FRAP values of extracts from *S. lucida, S. laevigata,* and *S. glauca*.

**Figure 4 plants-11-02793-f004:**
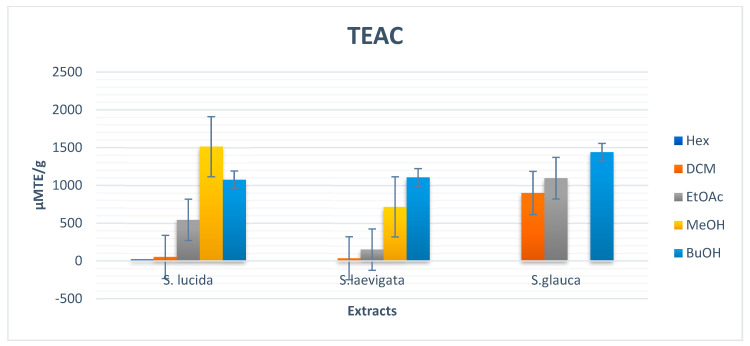
Bar chart of TEAC values of extracts from *S. lucida, S. laevigata,* and *S. glauca*.

**Table 1 plants-11-02793-t001:** Oxygen radical absorbance capacity (ORAC) activities of *S. lucida, S. laevigata,* and *S. glauca* extracts.

	ORAC (µM TE/g)		
Extracts	*S. lucida*	*S. laevigata*	*S. glauca*
Hexane	1013.65 ± 12.26	924.25 ± 17.77	NA *
Dichloromethane	1048.15 ± 35.35	1067.17 ± 38.95	3207.09 ± 79.34
Ethyl acetate	4010.56 ± 73.52	1363.86 ± 72.80	4574.93 ± 109.12
Methanol	5793.45 ± 27.30	3033.18 ± 222.16	NA *
Butanol	4198.42 ± 166.53	7475.11 ± 73.23	5653.36 ± 328.66
Trolox	14.2 ± 04.01		

Data presented as means ± SDs (standard deviations). NA * = not active at the tested concentrations.

**Table 2 plants-11-02793-t002:** Ferric ion reducing power (FRAP) and Trolox equivalent antioxidant capacity (TEAC) activities of *S. lucida, S. laevigata,* and *S. glauca* extracts.

	FRAP (µM AAE/g)			TEAC (µM TE/g)		
Extracts	*S. lucida*	*S. laevigata*	*S. glauca*	*S. lucida*	*S. laevigata*	*S. glauca*
Hexane	12.99 ± 1.82	20.52 ± 4.27	NA *	20.26 ± 4.42	NA *	NA *
Dichloromethane	40.69 ± 2.11	39.34 ± 6.94	431.62 ± 39.34	52.20 ± 2.82	35.77 ± 4.01	900.44 ± 49.14
Ethyl acetate	137.24 ± 16.54	86.28 ± 9.16	557.09 ± 39.41	543.24 ± 9.34	150.50 ± 12.68	1095.42 ± 28.42
Methanol	1038.39 ± 80.41	411.58 ± 79.73	NA *	1512.08 ± 0.06	715.66 ± 6.76	NA *
Butanol	680.01 ± 96.34	977.88 ± 71.24	1008.67 ± 41.65	1075.2 ± 114.89	1104.67 ± 24.61	1438.63 ± 13.53
Ascorbic acid	390.12 ± 09.12			-	-	-
Trolox	-	-	-	205.01 ± 05.03		

Data presented as means ± SDs (standard deviations). NA * = not active at the tested concentrations.

**Table 3 plants-11-02793-t003:** Inhibitory activity (%) of *Searsia* (*S. lucida, S. laevigata,* and *S. glauca*) extracts on some carbohydrate digestive enzymes (alpha glucosidase and alpha amylase).

	Alpha Glucosidase (%)			Alpha Amylase (%)		
Extracts	*S. lucida*	*S. laevigata*	*S. glauca*	*S. lucida*	*S. laevigata*	*S. glauca*
Hexane	91.85 ± 1.30	73.92 ± 2.30	59.13 ± 1.02	24.95 ± 2.12	41.81 ± 1.04	41.2 ± 1.23
Dichloromethane	49.28 ± 1.19	68.28 ± 1.12	85.22 ± 2.07	33.47 ± 1.21	40.99 ± 2.12	50.08 ±1.06
Ethyl acetate	93.54 ± 2.01	79.99 ± 1.23	62.50 ± 1.05	50.16 ± 1.71	52.31 ± 1.01	39.85 ± 1.41
Butanol	67.74 ± 1.27	90.10 ± 2.06	55.76 ± 1.45	60.88 ± 2.21	59.72 ± 2.14	47.71 ± 2.54
**Compounds**						
**IC_50_ (μg/mL)** values of tested compounds on alpha glucosidase and alpha amylase						
Amentoflavone	5.57 ± 1.17			19.84 ± 1.03		
Moronic acid	10.57 ± 2.02			20.08 ± 0.98		
Quercetin	105.41 ± 2.02					
Acarbose					10.25 ± 1.23	

Data presented as means ± SDs (standard deviations).

## Data Availability

Not applicable.

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
