# Peer review of "Phytochemical Investigation and Biological Studies on Selected Searsia Species"

_plants, 2022, doi:10.3390/plants11202793_

Round 1

Reviewer 1 Report

In the study "Phytochemical investigation and biological studies on selected Searsia species" , the authors studied the chemical structures of the isolated compounds from S. gluaca, S. Lucida and S. laevigata and antioxidant and alpha glucosidase assay. The study was interesting. However, the manuscript needs to be improved for publication in the Plants MDPI journal.

Comments

1) In Table 2 report Ferric ion reducing (FRAP) and Trolox equivalent antioxidant capacities (TEAC) of S. lucida, S. laevigata, and S. glauca extracts. Authors need to mention the statistical significance compared with the standard.

2 ) Mode of antidiabetic mechanism need to be discussed

3) Authors studied the isolated compounds using FTIR , HPLC, NMR...however, no graphical image / data found

4) The author should discuss about the ethnomedical importance and earlier studies conducted in Searsia glauca, Searsia Lucida, and Searsia laevigata 

Reviewer 2 Report

The article entitled "Phytochemical investigation and biological studies on selected Searsia species" is very well written and could be published in Plants after a little revision. In line 255, the word "by" is redundant

In line 263 it is not said what the column was finally washed with: "... and washed with 100%." of what, solvent with higher polarity?

Round 2

Reviewer 1 Report

Dear editor,

The authors responsed to the queries point by point and manuscript could be accepted for possible publication.